# Effect of Relative Humidity on the Composition of Secondary Organic Aerosol from Oxidation of Toluene

Mallory L. Hinks,[1] Julia Montoya-Aguilera,[1] Lucas Ellison,[1] Peng Lin,[2] Alexander Laskin,[2] Julia Laskin,[2] Manabu Shiraiwa,[1] Donald Dabdub,[3] and Sergey A. Nizkorodov[1]

[1]Department of Chemistry, University of California Irvine, Irvine, CA 92697
[2]Department of Chemistry, Purdue University, West Lafayette, IN 47907, USA
[3]Department of Mechanical and Aerospace Engineering, University of California Irvine, Irvine, CA 92697

*Correspondence to*: Sergey A. Nizkorodov (nizkorod@uci.edu)

**Abstract.** The effect of relative humidity (RH) on the chemical composition of secondary organic aerosol (SOA) formed from low-$NO_x$ toluene oxidation in the absence of seed particles was investigated. SOA samples were prepared in an aerosol smog chamber at <2% RH and 75% RH, collected on Teflon filters and analyzed with nanospray desorption electrospray ionization high-resolution mass spectrometry (nano-DESI-HRMS). Measurements revealed a significant reduction in the fraction of oligomers present in the SOA generated at 75% RH compared to SOA generated under dry conditions. In a separate set of experiments, the particle mass concentrations were measured with a Scanning Mobility Particle Sizer (SMPS) at RHs ranging from <2% to 90%. It was found that the particle mass loading decreased by nearly an order of magnitude when RH increased from <2% to 75-90% for low-$NO_x$ toluene SOA. The volatility distributions of the SOA compounds, estimated from the distribution of molecular formulas using the "molecular corridor" approach, confirmed that low-$NO_x$ toluene SOA became more volatile on average under high RH conditions. In contrast, the effect of RH on SOA mass loading was found to be much smaller for high-$NO_x$ toluene SOA. The observed increase in the oligomer fraction and particle mass loading under dry conditions were attributed to enhancement of condensation reactions, which produce water and oligomers from smaller compounds in low-$NO_x$ toluene SOA. The reduction in the fraction of oligomeric compounds under humid conditions is predicted to partly counteract the previously observed enhancement in the toluene SOA yield driven by the aerosol liquid water chemistry in deliquesced inorganic seed particles.

## 1 Introduction

Secondary organic aerosol (SOA) is an important component of atmospheric particulate matter. It is formed in the atmosphere via oxidation of volatile organic compounds (VOCs) by common atmospheric oxidants such as $O_3$, OH, and $NO_3$ (Seinfeld and Pandis, 2016). The SOA formation mechanisms depend in a complex way on physical environmental parameters such as solar irradiance, temperature, and relative humidity (RH). They also depend on the type of oxidant, concentration of VOC precursors, which govern $RO_2$ concentrations, and $NO_x$ levels, which determine the fate of the $RO_2$ radicals. The RH controls the amount of available water in the system, and therefore

affects processes in which water acts as a reactant, product, or solvent in several ways. Firstly, gaseous water can directly participate in the VOC oxidation reactions. For example, it is well known to react with carbonyl oxide intermediates in the ozonolysis of alkenes (Finlayson-Pitts and Pitts Jr, 2000). Additionally, aerosol liquid water (ALW) present in hygroscopic particles can lead to hydrolysis of organic compounds and other particle-phase reactions involving or catalyzed by water (Ervens et al., 2011). ALW also has a strong effect on the acidity of particles and, therefore, affects acid-catalyzed processes occurring in particles (Jang et al., 2002). Furthermore, water can act as a plasticizer for SOA particles making them less viscous, thus affecting the rate of their growth (Renbaum-Wolff et al., 2013; Perraud et al., 2012; Shiraiwa and Seinfeld, 2012). Under supersaturated conditions, aqueous chemistry occurring in cloud and fog droplets promotes conversion of small water-soluble molecules into non-volatile products that would not form in the absence of liquid water (Herrmann et al., 2015). Finally, dissolved SOA compounds may undergo more efficient photodegradation in water (Bateman et al., 2011; Nguyen et al., 2012; Romonosky et al., 2015; Romonosky et al., 2017; Zhao et al., 2017) compared to dry particles (Kourtchev et al., 2015).

Chemical composition is an important characteristic of SOA because it may determine the climate and health relevant properties of particles. The effect of RH on the chemical composition of particles has been studied for several types of biogenic SOA (Nguyen et al., 2011; Zhang et al., 2011; Riva et al., 2016; Harvey et al., 2016). For example, Nguyen et al. (2011) examined high-$NO_x$ isoprene SOA formed under high and low RH conditions and found that the high RH samples contained fewer oligomers than the low RH samples. Zhang et al. (2011) investigated the effect of RH on the composition of high-$NO_x$ isoprene SOA and found that oligoesters present in the SOA were suppressed at higher RH, while the formation of organosulfates was enhanced. Riva et al. (2016) studied the effect of RH on SOA made from oxidized isoprene hydroxyhydroperoxide (ISOPOOH) and found that increasing RH led to an increase in abundance of some oligomers while decreasing the abundance of other oligomers. Harvey et al. (2016) investigated the effect of RH on 3-hydroxypropanal ozonolysis SOA and found that increasing RH resulted in a decrease in SOA yield and a decrease in oligomerization.

The effect of RH on SOA formed from monoaromatic compounds such as benzene, toluene, m-xylene, and 1,3,5-trimethylbenzene (TMB) has been studied as well. Most of these studies focused on the effect of RH on SOA yield (Edney et al., 2000; Zhou et al., 2011; Cocker III et al., 2001; Kamens et al., 2011; Cao and Jang, 2010; Faust et al., 2017; Liu et al., 2017; Jia et al., 2017) and not chemical composition of SOA particles. The comparison between different experiments is complicated by the fact that some experiments are done in presence of hygroscopic seed particles, where ALW may be playing a role, others are done with seed particles containing strong acids, which favor acid catalyzed chemistry, and others use no seed particles. For toluene SOA produced in presence of hygroscopic seed particles, the yield is generally found to be larger under high RH conditions (Zhou et al., 2011; Kamens et al., 2011; Faust et al., 2017; Liu et al., 2017; Jia et al., 2017) because additional organic compounds are produced by aqueous photochemistry of small highly soluble compounds, such as glyoxal, partitioned in ALW. However, Cao and Jang (2010) observed a negative correlation between RH and SOA yield for low-$NO_x$ experiments, i.e., lower RH resulted in higher SOA yields.

In the experiments done without seed particles, the acid catalysis and chemistry occurring in ALW do not contribute to the particle growth. However, this does not rule out a possibility that RH may affect the SOA chemical composition and yield by other mechanisms mentioned above. Indeed, White et al. (2014) investigated the effect of RH on composition of toluene SOA produced under high-$NO_x$ conditions without seed particles and observed higher toluene SOA yields at elevated RH as well as higher yields of photooxidation products. In this work, we studied the composition of low-$NO_x$ toluene SOA formed under dry and humid conditions in the absence of seed particles. We observed a significant negative correlation between RH and low-$NO_x$ SOA from toluene SOA mass loading and a strong RH dependence on SOA molecular composition. We attribute this effect to the more extensive oligomerization of SOA compounds driven by condensation reactions under dry conditions. The reduction in the fraction of oligomeric compounds under humid conditions is predicted to partly counteract the previously observed enhancement in the toluene SOA yield driven by the ALW chemistry in deliquesced inorganic seed particles.

## 2 Materials and Methods

SOA was generated by photooxidation of toluene in a 5 $m^3$ smog chamber surrounded by a bank of UV-B lights. Before each experiment, the chamber was humidified to the desired RH by flowing purified air (typical VOC mixing ratios below 1 ppb) through a Nafion humidifier (PermaPure). The temperature ($\pm 1$ °C) and RH ($\pm 2\%$ RH) inside the chamber were monitored with a Vaisala HMT330 probe. No seed aerosol was used. Hydrogen peroxide ($H_2O_2$) was introduced to the chamber by injecting a measured volume of aqueous $H_2O_2$ (30 wt%) into a bulb where it was evaporated and carried into the chamber by a flow of purified air over a period of 30 minutes. The majority of the experiments were done under low-$NO_x$ conditions with concentrations of NO and $NO_y$ being below the 1 ppb detection limit of the $NO_y$ analyzer (Thermo Scientific 42i-Y). In the high-$NO_x$ experiments, gaseous NO (1000 ppm in $N_2$) was added to achieve a total NO concentration of 300 ppb. Toluene (Fisher Scientific, ACS grade) was introduced into the chamber by evaporating a measured volume of liquid toluene into a stream of air over a period of five minutes, which resulted in a toluene mixing ratio of either 300 ppb or 1000 ppb. Following the addition of gaseous reactants into the chamber, the UV lamps were turned on, photolyzing the $H_2O_2$ and resulting in a steady-state OH concentration of $1 \times 10^6$ molec·$cm^{-3}$ (determined in a separate experiment). These high concentrations of toluene were chosen in order to produce a sufficient amount of SOA to collect for offline analysis. Throughout each experiment, particle concentrations were monitored with a Scanning Mobility Particle Sizer (SMPS Model 3080, TSI Inc.). We used an effective SOA particle density of 1.4 g $cm^{-3}$ to convert the SMPS measurements into particle mass concentration (Sato et al., 2007; Ng et al., 2007). The concertation of toluene in the chamber was tracked with a proton-transfer reaction time of flight mass spectrometer (PTR-ToF-MS). The observed decrease in the toluene concentration (typically ~40%) was the same under low and high RH conditions.

SOA samples were collected onto Teflon filters for offline analysis by nano-DESI-HRMS. The filters were sealed and frozen immediately after the sample collection to avoid decomposition of less stable compounds, as observed for example by Krapf et al. (2016). The SOA filter samples were brought to room temperature and immediately analyzed in both positive and negative ion modes using an LTQ-Orbitrap mass spectrometer (Thermo Corp.) with a resolving power of $10^5$ at $m/z$ 400 equipped with a custom-built nano-DESI source (Roach et al., 2010a, b). The

advantage of nano-DESI is in minimizing the time in which the sample is exposed to the solvent, thus minimizing solvolysis reactions. Mass spectra of the solvent and blank filters were recorded as controls. Mass spectra of the samples with highest signal-to-noise ratio were clustered together, and the *m/z* axis was calibrated internally with respect to known SOA products. The solvent and impurity peaks were discarded. The peaks were assigned formulas,

$C_cH_hO_oN_nNa_{0-1}^+$ or $C_cH_hO_oN_n^-$, constrained by valence rules and elemental ratios (c,h,o,n refer to the number of corresponding atoms in the ion) (Kind and Fiehn, 2007). The resulting ion formulas were converted into formulas of the corresponding neutral species. All data reported below refers to the formula and molecular weights of the neutral species.

### 3 Results and Discussion

The mass spectra of a low RH sample (<2% RH) and a high RH sample (75% RH) are shown in Figure 1, plotted as a function of the molecular weight of the neutral compound. The mass spectra obtained in the positive and negative ion modes represent the SOA compounds ionizable in these modes, and are not expected to be identical (Walser et al., 2008). The low-$NO_x$ mass spectrum shown in Figure 1 is qualitatively similar to the low-$NO_x$ mass spectrum of toluene SOA discussed in a previous study where it was prepared in a different smog chamber but analyzed by the

same nano-DESI instrument (Lin et al., 2015).

As shown in Figure 1, the increase in RH resulted in a visible reduction in the overall peak abundance for both ion modes, due to the fact that the high RH sample had a much lower particle mass during the SOA generation (see below), and thus there was less material on the substrate. There was also a reduction in the number of observed peaks. For example, the positive mode mass spectrum in Figure 1 contains 665 peaks at low RH but only 285 peaks

at high RH; the corresponding peak numbers are 276 and 90 for the negative ion mode. Despite this reduction in peak abundance and number, the major observed peaks in the mass spectra remained the same. Table 1 lists five most abundant peaks for both the low and high RH samples observed in positive and negative modes. The fact that the major peaks are similar between the low and high RH samples suggests that the major products are produced by a similar mechanism that is not too sensitive to RH. It is of course still possible that the distribution of different

structural isomers within each peak could be affected by humidity but the nano-DESI method used here would be blind to this effect because it cannot separate isobaric species.

While the major oxidation products were similar at low and high RH, the less abundant products were much more strongly affected by RH. Specifically, the abundances of some high-molecular weight compounds were visibly reduced at high RH (Figure 1), suggesting that either the gas-phase oligomer formation is suppressed by water vapor

or the particle-phase oligomer formation is suppressed by ALW. (An alternative explanation is that oligomers hydrolyze after partitioning into the particle but the amount of ALW in the particles might be too small to sustain efficient hydrolysis). To better quantify this effect, Figure 2 shows the combined peak abundances as a function of the number of carbon atoms ($n_C$) in each molecule. Monomer compounds containing $n_C$=7 and dimer compounds with $n_C$=14 clearly dominate the distribution. In fact, the combined abundance of dimers ($n_C$=14) represents the

highest peak in the distribution in the positive ion mode. Many larger compounds with $n_C$ up to 32 also appear in the mass spectrum, and these minor compounds are the ones most affected by RH.

When comparing the low RH sample to the high RH sample, there is a significant decrease in combined peak abundance for molecules with $n_C>7$ under high RH conditions (except for the $n_C=14$ dimers). Because these higher molecular weight oligomers tend to have lower volatility (Li et al., 2016), they play an important role in the formation and growth of aerosol particles. With the lower fraction of oligomers produced under high RH conditions, the population of the oxidation products becomes more volatile on average, which should result in a lower SOA yield.

To better illustrate the possible effect of RH on the yield of condensable oxidation products, the volatility distributions were estimated for the low-$NO_x$ toluene SOA compounds using the "molecular corridor" approach (Li et al., 2016; Shiraiwa et al., 2014). This parameterization was developed specifically for atmospheric organic compounds containing oxygen, nitrogen, and sulfur (Li et al., 2016), and it makes it possible to estimate the pure compound vapor pressure, $C_0$, from the elemental composition derived from high-resolution mass spectra (Lin et al., 2016; Romonosky et al., 2017). $C_0$ is related to the more commonly used effective saturation mass concentration, $C^*$ = $\gamma \times C_0$, where $\gamma$ is the activity coefficient (Pankow, 1994). $C_0$ becomes equal to $C^*$ under the assumption of an ideal thermodynamic mixing. The $C_0$ values were calculated for each compound observed in the positive and negative ion mode mass spectra. The values were binned in equally spaced bins of base-10 logarithm of $C_0$ as is commonly done in the volatility basis set (VBS) (Donahue et al., 2006). The contribution of each compound to its volatility bin was taken to be proportional to its relative abundance in the mass spectrum. Because of the correlation between the ESI detection sensitivity and molecular weight (Nguyen et al., 2013), the mass fraction of the detected SOA compound is approximately proportional to its peak abundance. This is a considerable approximation because even for a series of carboxylic acids the ESI detection sensitivities can vary by several orders of magnitude within the same sample (Bateman et al., 2012). However, this approximation may still be useful for comparing distributions for the same types of SOA produced and analyzed under the same experimental conditions (Romonosky et al., 2017).

Figure 3 shows the resulting distribution of the SOA compounds by volatility. Under typical ambient conditions, compounds with $C_0$ above ~10 µg m$^{-3}$, i.e., the ones falling above the $\log(C_0) = 1$ bin, should exist primarily in the gaseous phase. Some of these more volatile compounds were detected in the negative ion mode. They may correspond to carboxylic acids that adsorbed to the filter during sampling. Less volatile compounds were preferentially observed in the positive ion mode. In both positive and negative ion modes, the compounds falling in the lower volatility bins were visibly suppressed at high RH. For example, the high RH to low RH ratio of the combined peak abundances for the compounds falling below $\log(C_0) = 1$ is 0.3 in the positive ion mode and 0.05 in the negative ion mode.

In order to investigate whether the decrease in oligomers affects the SOA mass loading, we have done additional experiments in which the particle mass concentration was tracked with SMPS at different RH. The SMPS data were corrected for particle wall loss effects assuming an effective first-order rate constant for the loss of mass concentration of $9.3 \times 10^{-4}$ min$^{-1}$, which was measured in a separate experiment (the rate constant was assumed to be independent of particle size). The SMPS experiments were performed under both low-$NO_x$ and high-$NO_x$ conditions. A summary of these experiments is presented in Table 2. Representative examples of the wall loss

corrected particle mass concentration as a function of photooxidation reaction time are shown in Figure 4 for both the low-NO$_x$ and the high-NO$_x$ toluene SOA systems.

Under high-NO$_x$ conditions, there was a small difference in the maximum mass concentration achieved under <2%, 40%, and 75% RH (less than a factor of 2), but under low-NO$_x$ conditions the difference was substantially larger. For the low-NO$_x$ system, the wall loss corrected particle mass concentration decreased by a factor of 8 over the range of RHs studied. The effect was reproducible as essentially the same mass concentration was observed in experiments repeated on different days under the same initial conditions.

Combining the measured particle mass concentrations with the toluene concentrations measurements from PTR-ToF-MS makes it possible to estimate the apparent SOA yields, which are listed in the last column of Table 2. Under high-NO$_x$ conditions, the yield decreased from ~27% to ~19% as RH increased from <2% to 77%. Under low-NO$_x$ conditions, the yields dropped from 15% to 2% for the same change in RH. We note the previously reported SOA yield from toluene SOA formed in the presence of ammonium sulfate seed aerosol displayed the opposite trend, with the yield being higher (~30%) under low-NO$_x$ conditions and lower under high-NO$_x$ conditions (~19 %) (Ng et al., 2007). Hildebrandt et al. (2009) noted that the yields in the toluene SOA system are highly sensitive to the oxidation conditions, including the type of UV lights used in photooxidation and the seed aerosol concentration. Furthermore, the wall loss effects are especially prominent in the toluene SOA system (Zhang et al., 2014). We attribute the difference in absolute values of yields between our experiments and experiments by Ng et al. (2007) to the difference in the experimental design.

The differences between the low and high RH systems cannot be explained by hygroscopic growth of particles at elevated RH. Throughout the experiment, the SMPS sampled air directly from the chamber. Each experiment lasted many hours, which allowed the sheath flow in the SMPS to approach the RH of the chamber air. Therefore, the particles sized by the SMPS contained some ALW and would appear larger than their dry size. If the organic mass in particles did not change at different RH levels, we would have observed an *increase* as opposed to a decrease in the measured particle mass concentration. With a typical hygroscopic growth factor (the ratio of particle diameters in humidified and dry air) for SOA of 1.1 at 85% RH (Varutbangkul et al., 2006), the increase in the apparent mass concentration would have been by a factor of about 1.3. Instead, the mass concentration decreased by almost a factor of 8 at higher RHs. The strong dependence of the low-NO$_x$ toluene SOA mass loading on RH is therefore not an artifact of the SMPS measurements.

We cannot rule out the possibility that the mass loading of SOA was affected by the enhanced wall loss of more water soluble compounds under high RH conditions. Indeed, the chamber wall effects are expected to be stronger for the slowly reacting toluene compared to monoterpenes that are oxidized much faster (Pierce et al., 2008). Furthermore, in the absence of seed particles, toluene SOA aerosol growth takes longer, making the wall loss effects larger (Kroll et al., 2007; Zhang et al., 2014). It is conceivable that the products of low-NO$_x$ oxidation of toluene are more water-soluble than the products of high-NO$_x$ oxidation of toluene. This would result in a stronger effect of RH on the mass loading of low-NO$_x$ SOA because these products would be more efficiently absorbed by the wetted chamber walls. Distinguishing the wall-loss effects from the effect of water on the distribution of oligomers would

require more careful chamber measurements of SOA yields over a broad range of concentrations and in the presence of seed aerosol (to suppress the wall loss effects).

A possible chemical explanation for the observed RH effect is that there are chemical reactions in the system that directly involve water and change the chemical composition of the particles, thereby affecting their growth rate. Previous studies have shown that RH can affect the composition and potential yield of SOA by altering the fraction of low-volatility oligomers in SOA. Increased RH could suppress oligomerization occurring by condensation reactions (i.e., reactions between monomers that produce water as a byproduct) by shifting the reaction equilibrium toward the products as discussed in Nguyen et al. (2011). Conversely, increased RH could promote hydrolysis of oligomers after they are produced in the gas-phase and partition into wet particles. As pointed out above, the latter mechanism is less likely due to the low ALW content of the organic particles.

To investigate the mechanism, we examined the frequency of occurrence of mass differences between the peaks in the high resolution mass spectra. Table 3 lists the most common mass differences in all four mass spectra. The most frequently observed mass difference in the low-RH sample was $C_2H_2O$, and its frequency of occurrence dropped in the high-RH sample. It is possible that $C_2H_2O$ results from oligomerization chemistry of glycolaldehyde ($C_2H_4O_2$), which can react by an aldol condensation mechanism with compounds containing a carbonyl group (Scheme 1). Glycolaldehyde has been observed previously in the oxidation of toluene (White et al., 2014; Yu et al., 1997), likely as an oxidation product of methylglyoxal. The mass difference corresponding to $H_2O$ was not amongst the most common, however, it became more probable in the high RH sample, consistent with hydration reactions. Anhydrides, commonly found in toluene SOA (Bloss et al., 2005; Forstner et al., 1997; Sato et al., 2007), may undergo hydrolysis, which adds an $H_2O$ unit to the formula.

We additionally tested whether oligomeric compounds occurring in low RH toluene SOA can be produced by either simple addition or condensation of monomer compounds occurring in high RH toluene SOA. If simple addition is responsible for the oligomerization, we would expect to see peaks in the low RH mass spectrum with molecular weights equal to the sum of two peaks from the high RH mass spectrum. If condensation is responsible for the oligomerization, we would expect to see peaks in the low RH mass spectrum with molecular weights equal to the sum of two peaks from the high RH mass spectrum minus the mass of water (the same relationship would hold in reverse for oligomers undergoing hydrolysis in the particle). In positive ion mode, the fraction of peaks that could be matched by the addition reactions was 69%, while the fraction of peaks matched by the condensation reactions was 83%. These numbers were 62% and 69%, respectively, for negative ion mode. This suggests that condensation reactions (that remove water) are more likely to be responsible for the enhanced oligomer formation under dry conditions. This conclusion is similar to the one reached in the study of the effect of RH on oligomerization in high-$NO_x$ isoprene SOA (Nguyen et al., 2011).

**4 Conclusions**

This study has demonstrated that the composition of low $NO_x$ toluene SOA depends on RH when it is produced in smog chamber experiments without seed particles. Oligomers produced by condensation reactions were observed in higher concentrations in the mass spectra of toluene SOA produced under low RH, and were suppressed under high

RH conditions. Additionally, the mass loading of low $NO_x$ toluene SOA was reduced under high RH conditions. The plausible reason for the suppression of SOA mass loading at high RH is the change in the SOA chemical composition that favors lower molecular weight, more volatile compounds. The reduction of dimers and trimers in the high RH samples suggests that low volatility oligomers are not forming in toluene SOA under low-$NO_x$

conditions, which means particle growth is suppressed and mass loading is reduced.

In previous studies on the effect of RH on SOA yield from toluene in the presence of hygroscopic seed (Zhou et al., 2011; Kamens et al., 2011; Faust et al., 2017; Liu et al., 2017; Jia et al., 2017), an opposite effect was observed in which the SOA yield increased at high RH. This was attributed to the aqueous partitioning and subsequent reactions in ALW of smaller photooxidation products, such as glyoxal (Faust et al., 2017). Our results suggest that the

increase in the SOA yield due to the ALW-supported chemistry is at least partly counteracted by the lower yield of oligomers under high RH conditions. While it is not straightforward to compare experiments done with and without seed aerosol in different smog chambers, the ALW-supported chemistry enhancement of the yield appears to be a more important effect.

It is conceivable that the effect of RH on the SOA yield is a common feature of all low-$NO_x$ aromatic SOA, all of

which should contain aldehyde compounds capable of oligomerization by condensation reactions. If this is the case, the production of SOA from naturally emitted aromatic compounds (indole, benzyl acetate, benzaldehyde, etc.), which exist in low-$NO_x$ environments, would be strongly modulated by the ambient relative humidity. This definitively warrants further investigation.

**Acknowledgements**

This publication was developed under Assistance Agreement No. 83588101 awarded by the U.S. Environmental Protection Agency to the Regents of the University of California. It has not been formally reviewed by EPA. The views expressed in this document are solely those of the authors and do not necessarily reflect those of the Agency. EPA does not endorse any products or commercial services mentioned in this publication. J. Montoya-Aguilera acknowledges support from the California LSAMP Bridge to the Doctorate Program at the University of California,

Irvine, which is funded by grant NSF-1500284. M. Shiraiwa acknowledges support from the National Science Foundation grant AGS-1654104. M. Shiraiwa and S.A. Nizkorodov acknowledge support from the Department of Energy grant DE-SC0018349. The PTR-ToF-MS instrument used in this work was purchased with grant NSF MRI-0923323. The HRMS measurements were performed at the W.R. Wiley Environmental Molecular Sciences Laboratory (EMSL) – a national scientific user facility located at PNNL, and sponsored by the Office of Biological

and Environmental Research of the U.S. DOE. PNNL is operated for U.S. DOE by Battelle Memorial Institute under Contract No. DE-AC06-76RL0 1830.

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

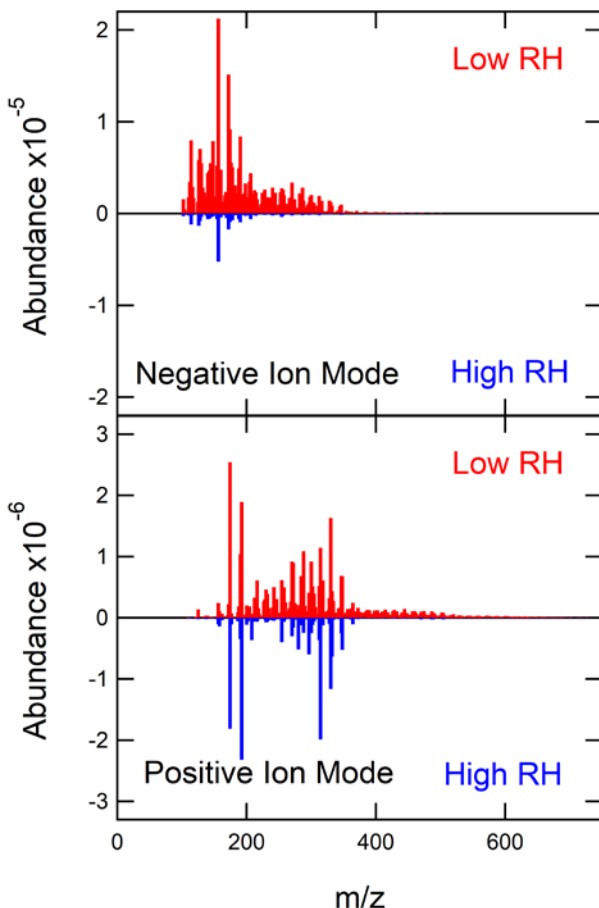

**Figure 1: High-resolution mass spectra obtained in negative ion mode (top) and positive ion mode (bottom). The red upward-pointing mass spectra represent the low-NO$_x$ SOA sample made under low RH (<2%) and the blue inverted mass spectra represent the low-NO$_x$ SOA sample made under high RH (75%).**

**Table 1: Five most abundant compounds observed in the low and high RH low-NO$_x$ toluene SOA samples. In positive ion mode, the most abundant species differed by one compound in the low and high RH experiments, hence the table contains 6 formulas. In negative ion mode, the same five most abundant peaks were observed at the low and high RH.**

| Positive Ion Mode | | Normalized Peak Abundance | |
|---|---|---|---|
| Nominal Mass | Formula | Low RH | High RH |
| 174 | $C_7H_{10}O_5$ | 1 | 1 |
| 192 | $C_7H_{12}O_6$ | 0.74 | 0.86 |
| 330 | $C_{14}H_{18}O_9$ | 0.64 | 0.78 |
| 314 | $C_{14}H_{18}O_8$ | 0.45 | 0.50 |
| 288 | $C_{12}H_{16}O_8$ | 0.43 | 0.11 |
| 332 | $C_{14}H_{20}O_9$ | 0.17 | 0.27 |
| Negative Ion Mode | | Normalized Peak Abundance | |
| Nominal Mass | Formula | Low RH | High RH |
| 156 | $C_7H_8O_4$ | 1 | 1 |
| 172 | $C_7H_8O_5$ | 0.71 | 0.86 |
| 174 | $C_7H_{10}O_5$ | 0.43 | 0.78 |
| 190 | $C_7H_{10}O_6$ | 0.40 | 0.50 |
| 114 | $C_5H_6O_3$ | 0.38 | 0.27 |

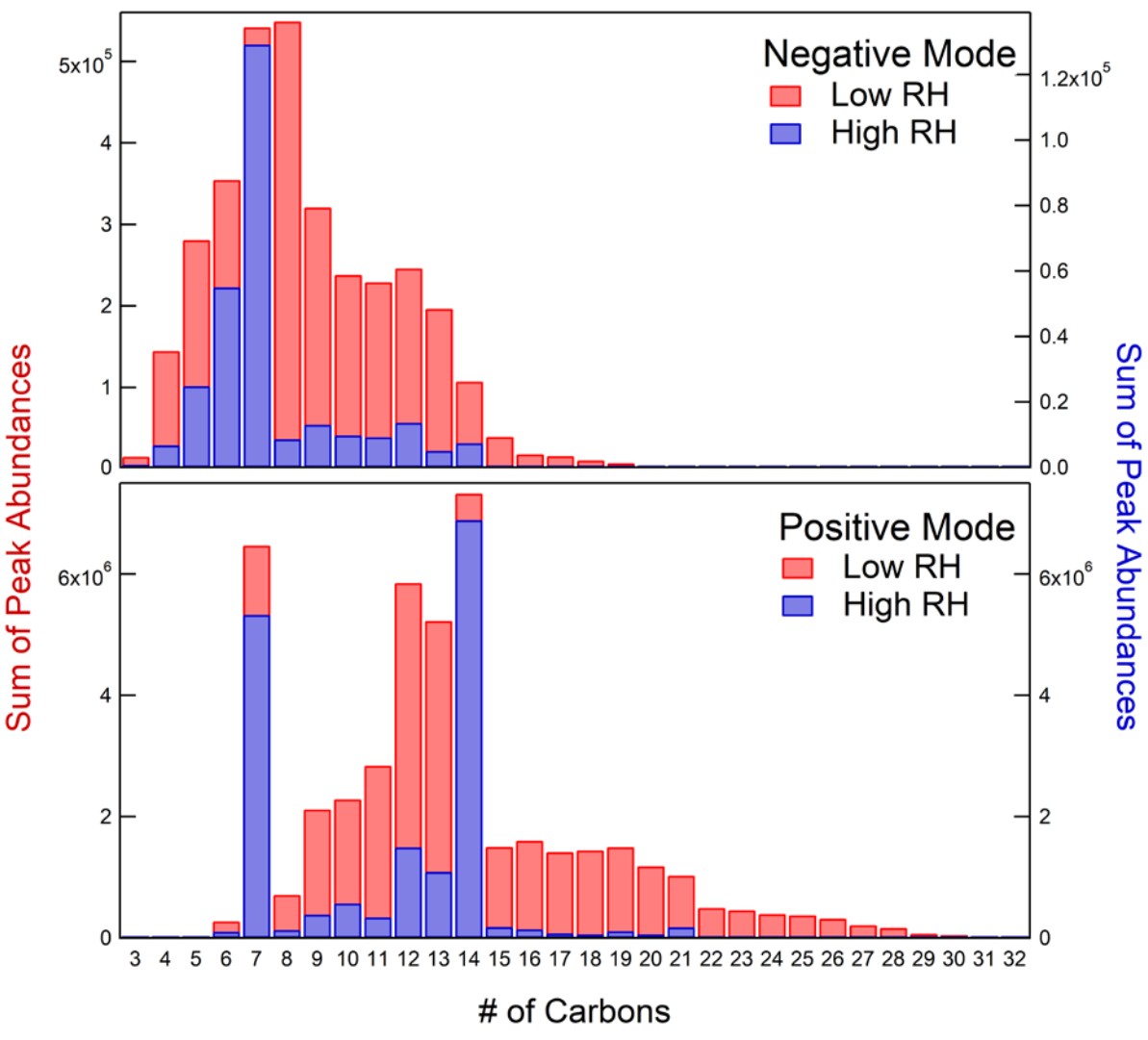

**Figure 2: Combined abundance of all peaks as a function of number of carbon atoms in negative mode (top) and positive mode (bottom). The data for the low RH sample are shown in red and the data for the high RH sample are shown in blue. The samples were prepared under low-NO$_x$ conditions.**

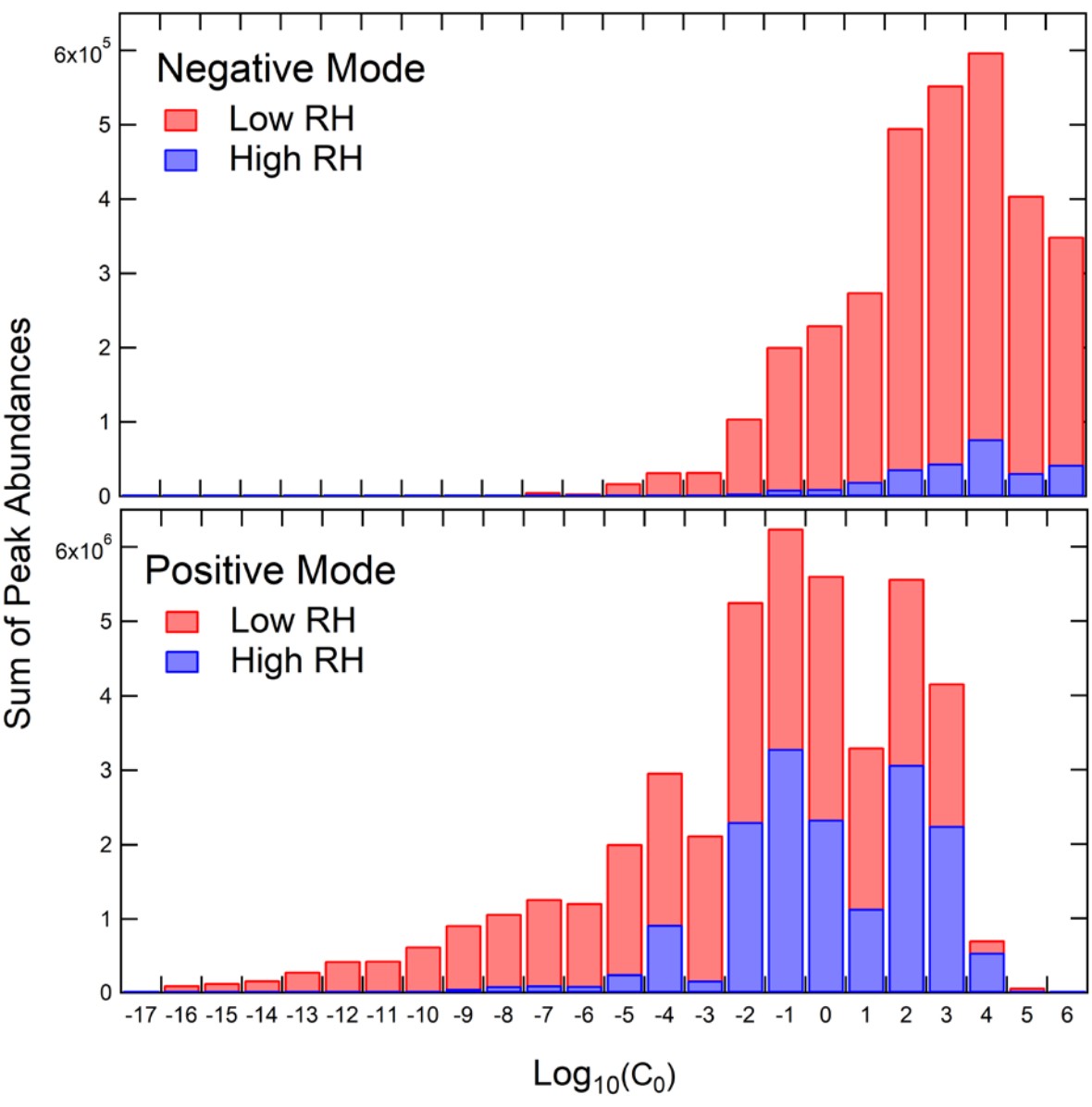

**Figure 3. Estimated volatility distribution for the compounds observed in low-NO$_x$ SOA samples in the negative (top) and positive (bottom) ion mode at high (blue bars) and low (red bars) RH. The height of each bar is proportional to the total ESI abundance of compounds falling within the volatility bin.**

**Table 2: Summary of SMPS experiments. The uncertainties included in this table are based on one standard deviation in the data for repeated experiments.**

| Initial RH | # of Experiments | NO$_x$ ppm | Toluene ppm | H$_2$O$_2$ ppm | SOA from SMPS µg/m$^3$ | Wall Loss Corrected SOA µg/m$^3$ | SOA Yield (%) |
|---|---|---|---|---|---|---|---|
| Low-NO$_x$ – High Toluene | | | | | | | |
| <2 | 4 | - | 1.0 | 2.0 | 180 ± 20 | 210 ± 20 | 15±2 |
| 20 ± 3 | 2 | - | 1.0 | 2.0 | 76 ± 4 | 87 ± 6 | 6.2±0.5 |
| 43 | 1 | - | 1.0 | 2.0 | 74 | 84 | 5.9 |
| 76 ± 1 | 4 | - | 1.0 | 2.0 | 27 ± 7 | 28 ± 7 | 2.0±0.5 |
| 89 ± 1 | 2 | - | 1.0 | 2.0 | 25 ± 8 | 26 ± 9 | 1.9±0.6 |
| Low-NO$_x$ – Lower Toluene | | | | | | | |
| <2 | 1 | - | 0.3 | 0.6 | 23 | 27 | 5.5 |
| 75 | 1 | - | 0.3 | 0.6 | 8 | 9 | 2.2 |
| High-NO$_x$ – High Toluene | | | | | | | |
| <2 | 1 | 0.3 | 1.0 | 2.0 | 330 | 390 | 27 |
| 43 | 1 | 0.3 | 1.0 | 2.0 | 210 | 260 | 18 |
| 77 | 1 | 0.3 | 1.0 | 2.0 | 230 | 270 | 19 |

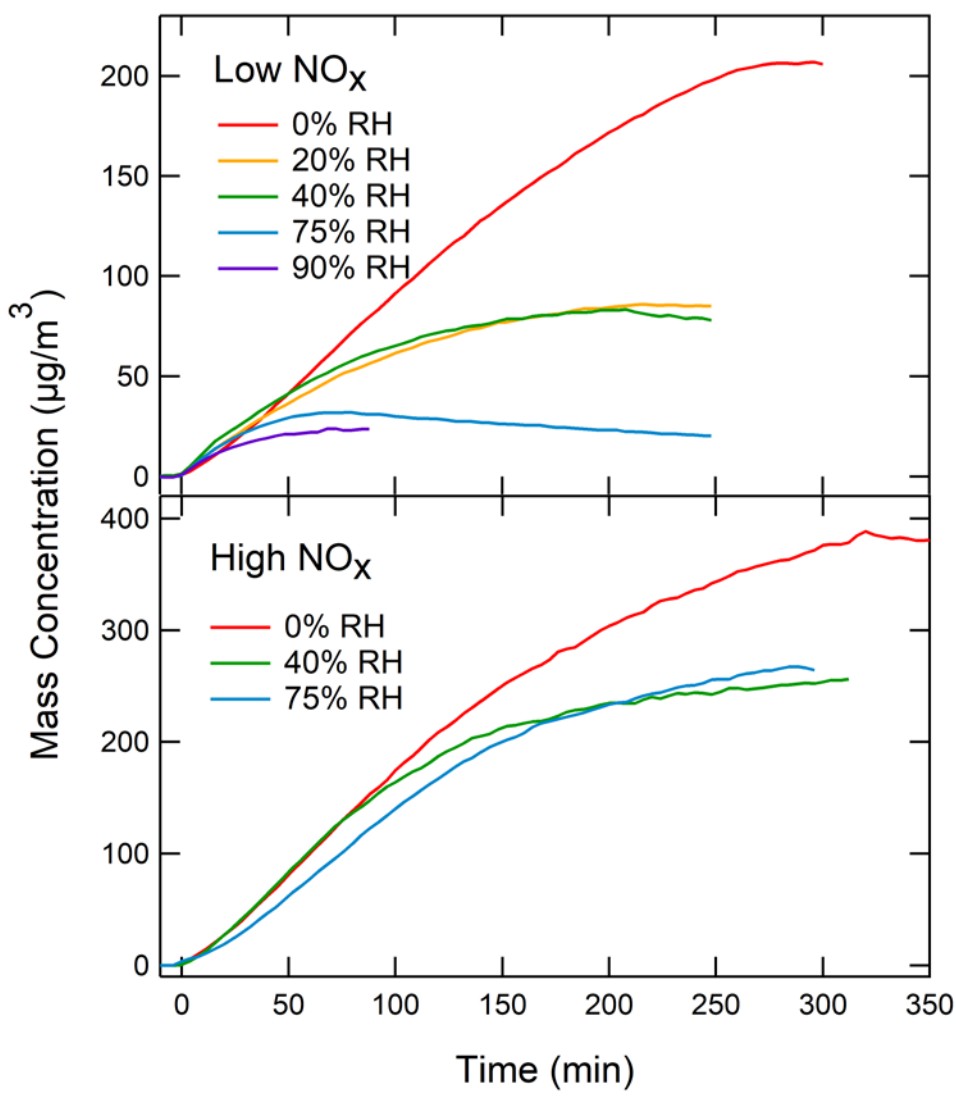

**Figure 4: Examples of particle mass concentration measurements by SMPS (corrected for wall-loss) as a function of photooxidation time under low-NO$_x$ (top) and high-NO$_x$ (bottom) conditions.**

**Scheme 1: An aldol condensation reaction involving glycolaldehyde that results in an addition of $C_2H_2O$ to the formula of the aldehyde co-reactant.**

**Table 3: Most common mass differences in the high resolution mass spectra of low-$NO_x$ toluene SOA (in the order of decreasing frequency of occurrence).**

| Positive Ion Mode | | Negative Ion Mode | |
|:---:|:---:|:---:|:---:|
| **Low RH** | **High RH** | **Low RH** | **High RH** |
| $C_2H_2O$ | $CH_2$ | C | O |
| $CH_2O$ | O | $C_2H_2O$ | $CH_2O$ |
| C | $CH_2O$ | O | C |
| $CH_2$ | $C_2H_2O$ | $CH_2O$ | $C_2H_2O$ |
| O | C | $CH_2$ | $CH_2$ |
| $C_3H_4O_2$ | CO | CO | $C_2H_2$ |