# Peer review of "Effect of Relative Humidity on the Composition of Secondary Organic Aerosol from Oxidation of Toluene"

_Atmospheric Chemistry and Physics, 2017_

## Referee Comment (RC1) · Anonymous Referee #1 · 21 Sep 2017

Review on:

**Hinks et al.: "Effect of Relative Humidity on the Composition of Secondary Organic Aerosol from Oxidation of Toluene"**

*General comments*

The authors describe a laboratory study on the atmospheric oxidation and secondary organic aerosol formation from toluene oxidation under different relative humidity conditions. Humidity dependence of the SOA composition is reported for low NOx conditions. The chemical composition of SOA collected on filter samples was analyzed by offline nano desorption electrospray ionization coupled to a high resolution mass spectrometer (nano-DESI-HRMS) in both polarity modes. The mass concentration of the formed SOA was monitored for a different set of chamber experiments using a scanning mobility particle sizer (SMPS). The authors report different chemical composition at different RH, and hypothesize that oligomerization by condensation reactions in the condensed phase can explain the observation of enhanced SOA yield at low humidity. In agreement with earlier studies they see no humidity dependence of toluene SOA formation under high NOx conditions.

The work addresses the relevant scientific question of anthropogenic SOA formation, and therefore contributes to build a better mechanistic understanding on urban SOA burden under different atmospheric conditions. Therefore the paper certainly lies within the scope of "Atmospheric Chemistry and Physics". It contributes with new experimental results on the SOA composition from toluene. As the authors state, toluene SOA yield experiments have already been described by others, but the linkage between SOA mass and composition has been missing. The scientific method of the SOA chamber experiment is clearly described and the measured SOA mass is corrected for chamber wall-losses. The offline analysis of Teflon filters is based on the established tool nano-DESI-HRMS, which provides the elemental composition of the condensed phase oxidation products as a basis for the hypothesis of different oligomerization efficiencies under different RH conditions.

I therefore recommend the article to be published in ACP, after the following specific comments and technical corrections have been addressed.

*Specific comments*

p.1 l.28: SOA formation mechanisms not only depend on solar irradiance, temperature and RH, but also on VOC precursor mixtures that ultimately govern RO2 radical concentration and on NOx levels which can affect branching and termination of oxidation pathways.

p.2 l.9: Another more recent publication (Zhao et al., 2017) that addresses a photochemical aqueous phase sink of dimers from VOC oxidation should be added.

p.3 l.16: A commonly known risk in offline analysis is (positive and negative) artefact formation on the filter, especially when reactive species, such as organic hydroperoxides (ROOH), organic peroxides (ROOR), peroxy acids etc., are sampled. This implies that the

measured composition not necessarily reflects the composition and state of the aerosol in the chamber. Krapf et al. (2017) have shown that a significant fraction of highly oxidized and thermodynamically unstable products of VOC ozonolysis can decompose on the scale of minutes to a few hours. This brings me to the question of how the authors evaluated the effect of filter artefacts? How long and under which conditions were the filters stored until analysis by nano-DESI-MS? Was the aerosol in the high RH experiments dried before being sampled on the filter? How and by which solvent were the filters extracted during nano-DESI-MS analysis?

p.4 l.1: The authors use the term "SOA compounds" when discussing mass spectral patterns. How was the possibility of ion source cluster formation evaluated?

p.4 l.4: The fact that common major peaks were observed, suggests that these are the same products, however, the possibility exists that different isomers are formed. Whether this is the case can only be evaluated by a separation technique before mass spectrometry, and should be mentioned in this paragraph.

p.4. l.7: The authors argue that oligomer formation (in the condensed phase) is suppressed under high RH condition. Have the authors considered that oligomerization (especially dimerization) can also occur in the gas phase by RO2-RO2 recombination? Once the dimers that are formed in the gas phase partition to the condensed phase, they might undergo more rapidly hydrolysis reactions under high RH conditions, which in the end would result in a similar observation.

p.4 l.27: ESI sensitivity is not only driven by mass but also by functionality, solvent composition, polarizability and by the tendency of the analyte to interact with other matrix components. Therefore it is highly questionable that peak abundance is proportional to the observed mass fraction. The strongest argument against the proposed simplified relationship between sensitivity and molecular weight is actually given in figure 1: If the sensitivity would only be a function of molecular weight, why is the monomer-to-dimer ratio different between the positive and negative mode? Furthermore, we observed that ESI sensitivity between two very similar commercially available standards (pinonic acid and camphoric acid) can be different by orders of magnitude. Therefore, the authors should argue here more carefully and consider that different ionization efficiency can introduce a significant bias.

p.5, l.1: I agree that the signal in the lower volatility bin (Figure 3) is suppressed under high RH in the positive mode. However, for the negative mode this is not obvious. Did the authors try to normalize the figures on the base peak in the mass spectrum to present the relative changes more in detail?

p.6, l.22-24: Again: the authors should discuss the possibility of enhanced condensed phase hydrolysis reactions of dimers under high RH conditions. The enhanced viscosity of particles under low RH might suppress the rate of hydrolysis reactions.

p.13, Figure 3: The volatility distribution in (a) looks as it is cut on the high volatility end. How many products appear in the bins $\log_{10}(C0)>6$? Can these products be summed up into one bin which contains all compounds of all bins with $\log_{10}(C0)>6$?

*Technical corrections*

p.13, Figure 3: The figure caption contains (a) and (b) which is not shown in the figure. Also, the colour description in the caption is wrong: high RH should be "blue" and low RH "red".

*Literature*

Krapf, M., El Haddad, I., Bruns, E. A., Molteni, U., Daellenbach, K. R., Prévôt, A. S. H., Baltensperger, U. and Dommen, J.: Labile Peroxides in Secondary Organic Aerosol, Chem, 1(4), 603–616, doi:10.1016/j.chempr.2016.09.007, 2017.

Zhao, R., Aljawhary, D., Lee, A. K. Y. and Abbatt, J. P. D.: Rapid Aqueous-Phase Photooxidation of Dimers in the α-Pinene Secondary Organic Aerosol, Environ. Sci. Technol. Lett., 4(6), 205–210, doi:10.1021/acs.estlett.7b00148, 2017.

---

## Referee Comment (RC2) · Anonymous Referee #2 · 22 Sep 2017

The Hinks et al. manuscript reports on differences in secondary organic aerosol (SOA) mass concentration and composition during high vs. low relative humidity (RH) chamber experiments. Toluene was oxidized under low-NOx and high-NOx and low (<2%) and high RH (up to 90%) conditions. Differences in mass concentration were evaluated using an SMPS; and differences in composition using nano-DESI-HRMS. While chamber experiments and SOA formation studies largely have been conducted under dry conditions, much recent attention has been given to the effects of water on particle formation and composition. Much of that effort has been focused on understanding the effects of RH on particle viscosity. There is much opportunity to advance the mechanistic understanding of SOA formation through compositional studies such as the one

presented in the Hinks et al. manuscript. That said, while the methodology and results are presented clearly, the analysis and discussion could be more robust. There have been a large number of papers published on the photooxidation of toluene (and other aromatics) and subsequent SOA formation. The results of the experiments presented are not adequately placed in the context of what is already known about toluene SOA formation, including recent mechanistic studies looking at the role of aerosol liquid water on SOA formation from toluene. The results presented are new, and with some further analysis and discussion, this work could become a significant contribution to the field. This work is suitable for publication in ACP, following some strengthening of the analysis and discussion.

Technical and Editorial Comments:

The abstract reports that the nano-DESI-HRMS analysis was performed on filters from the low (<2%) and high (75% only) experiments, and particle size analysis on a wider range of high RH (75-90%). This is not as clear in the manuscript itself. Are the nano-DESI-HRMS results from a single experiment? Or averaged over all of the 75% RH experiments? This should be made clearer in the text and in the figures/tables.

p. 4, lines 1-4 (referencing Fig. 1/Table 1): The observation that there seems to be no mechanistic difference between high and low RH and only differences in the formation of oligomers is not completely satisfactory. For the positive ion mode, it is clear that the mass spectra are more similar (samples share the same "compounds"); it is not quite as clear that there is a reduction in peak abundance. For the negative ion mode, it is clearer that there is a reduction in peak abundance, but it is not as clear that there is similarity among the most abundant compounds. For both positive and negative mode there seems to be a reduction in peak diversity, with significant differences in the negative ion mode.

Can these differences be further probed to support the hypothesis or provide alternative hypotheses?

[Figure]

There have been a number of mechanistic/product studies of toluene, and other aromatics, under high- and low-NOx conditions. Do those studies (e.g., observed gas-phase intermediates) combined with what is known about oligomerization pathways (e.g., as discussed on p.6) support the RH dependence under low NOx but not high NOx conditions? Citation of similar results in Cao et al. (2010) is not sufficient.

p. 4, lines 5-10 (referencing Fig. 2): While the formation of higher order oligomers is suppressed, it is interesting that the most abundant peak > C7 in the positive ion mode spectra under high RH appears to be a dimer.

p. 4, lines 31-34: The fraction of compound detected in the particle phase is discussed in terms of ambient organic aerosol (OA) levels, and an explanation is given for gas-phase adsorption artifacts. However, based on table 2 (and discussion in the manuscript) the experimental OA levels vary widely between the low and high RH experiments. The discussion of partitioning and artifacts needs to be expanded to reflect the range of experimentally observed OA levels. One question that arises is whether there are more artifacts with high RH than low RH and if so what may be the reason for that?

There is little to no discussion of the role gas-phase chemistry plays in the observations. There is an underlying assumption that because the initial VOC and oxidant concentrations were the same between experiments that the reacted VOC concentrations were also the same. Are there measurements to support that assumption? While many things can influence SOA yield, were the observed yields (if known) generally consistent with other published studies?

Did the average particle size change between high and low NOx conditions?

p. 5, lines 17-27: The authors mention the hygroscopicity of SOA here, but further discussion may be helpful. Is the amount of aerosol liquid water under high RH (75%) sufficient to prohibit/limit condensation reactions?
p. 7, lines 3-12: The extension of the experimental observations to atmospheric implications is not well supported and unnecessary. There are so many elements of experimental design that may affect the results and the extension to the ambient atmosphere; these include absence of seed aerosol (chemical and physical effects), absolute levels of precursors and particles, and relative levels of radicals. Further in the ambient atmosphere the gas-phase chemistry is controlled by more than a single VOC precursor, and the particle composition will affect the extent of aerosol liquid water.

Figures: It is recommended to specify the NOx and RH conditions in the figure captions just as a reminder since high-NOx and a range of RH are discussed in the manuscript.

---

## Author Comment (AC1) · 26 Dec 2017

**Response to Reviewer 1**

General comments

The authors describe a laboratory study on the atmospheric oxidation and secondary organic aerosol formation from toluene oxidation under different relative humidity conditions. Humidity dependence of the SOA composition is reported for low NOx conditions. The chemical composition of SOA collected on filter samples was analyzed by offline nano desorption electrospray ionization coupled to a high resolution mass spectrometer (nano-DESI-HRMS) in both polarity modes. The mass concentration of the formed SOA was monitored for a different set of chamber experiments using a scanning mobility particle sizer (SMPS). The authors report different chemical composition at different RH, and hypothesize that oligomerization by condensation reactions in the condensed phase can explain the observation of enhanced SOA yield at low humidity. In agreement with earlier studies they see no humidity dependence of toluene SOA formation under high NOx conditions.

The work addresses the relevant scientific question of anthropogenic SOA formation, and therefore contributes to build a better mechanistic understanding on urban SOA burden under different atmospheric conditions. Therefore the paper certainly lies within the scope of "Atmospheric Chemistry and Physics". It contributes with new experimental results on the SOA composition from toluene. As the authors state, toluene SOA yield experiments have already been described by others, but the linkage between SOA mass and composition has been missing. The scientific method of the SOA chamber experiment is clearly described and the measured SOA mass is corrected for chamber wall-losses. The offline analysis of Teflon filters is based on the established tool nano-DESI-HRMS, which provides the elemental composition of the condensed phase oxidation products as a basis for the hypothesis of different oligomerization efficiencies under different RH conditions.

I therefore recommend the article to be published in ACP, after the following specific comments and technical corrections have been addressed.

Specific comments

**1.1** p.1 l.28: SOA formation mechanisms not only depend on solar irradiance, temperature and RH, but also on VOC precursor mixtures that ultimately govern RO2 radical concentration and on NOx levels which can affect branching and termination of oxidation pathways.

Thank you for pointing this out. A note about the importance of RO2 concentrations and NOx levels has been added to the introduction.

**1.2** p.2 l.9: Another more recent publication (Zhao et al., 2017) that addresses a photochemical aqueous phase sink of dimers from VOC oxidation should be added.

This reference has been included in the text, as well as a few other related references.

**1.3** p.3 l.16: A commonly known risk in offline analysis is (positive and negative) artefact formation on the filter, especially when reactive species, such as organic hydroperoxides (ROOH), organic peroxides (ROOR), peroxyacids etc., are sampled. This implies that the measured composition not necessarily reflects the composition and state of the aerosol in the chamber. Krapf et al. (2017) have shown that a significant fraction of highly oxidized and thermodynamically unstable products of VOC ozonolysis can decompose on the scale of minutes to a few hours. This brings me to the question of how the authors evaluated the effect of filter artefacts? How long and under which conditions were the filters stored until analysis by nano-DESI-MS? Was the aerosol in the high RH experiments dried before being sampled on the filter? How and by which solvent were the filters extracted during nano-DESI-MS analysis?

This is a good point. In order to avoid decomposition of organic compounds, the filter was placed inside a vacuum sealed pouch and frozen immediately after the collection. It was then shipped in dry ice overnight to PNNL for HRMS analysis. The aerosol in the high RH experiments was not dried during the collection, and it is possible that the amount of decomposition occurring on the filter during collection was different in high vs. low RH experiments. We have not corrected for this filter artefact.

We should clarify that the filters were never extracted. The advantage of nano-DESI is that it dissolves the sample on the fly and analyzes it within seconds of exposure of the sample to the solvent. We have added a note about this to the last paragraph in the Materials and Methods section.

**1.4** p.4 l.1: The authors use the term "SOA compounds" when discussing mass spectral patterns. How was the possibility of ion source cluster formation evaluated?

In our previous work we observed that weakly bound clusters do not survive the injection of ion into the Orbitrap. Therefore, we have not examined the possibility of contamination of the mass spectrum with cluster ions in this study. We have not made a change to the paper in response to this comment.

**1.5** p.4 l.4: The fact that common major peaks were observed, suggests that these are the same products, however, the possibility exists that different isomers are formed. Whether this is the case can only be evaluated by a separation technique before mass spectrometry, and should be mentioned in this paragraph.

We agree with this assessment, and have included a note about this at the end of the second paragraph in the Results and Discussion section.

**1.6** p.4. l.7: The authors argue that oligomer formation (in the condensed phase) is **suppressed** under high RH condition. Have the authors considered that oligomerization (especially dimerization) can also occur in the gas phase by RO2-RO2 recombination? Once the dimers that are formed in the gas phase partition to the condensed phase, they might undergo more rapidly hydrolysis reactions under high RH conditions, which in the end would result in a similar observation.

The reviewer is correct in recognizing that we cannot distinguish between the two scenarios in which oligomer formation is suppressed by high RH or oligomers are destroyed more efficiency at high RH. We have added both possibilities in the text in the third paragraph in the Results and Discussion section. However, we pointed out that the amount of water in particles is too small and the possibility of hydrolysis is less likely than suppression of condensation.

**1.7** p.4 l.27: ESI sensitivity is not only driven by mass but also by functionality, solvent composition, polarizability and by the tendency of the analyte to interact

with other matrix components. Therefore it is highly questionable that peak abundance is proportional to the observed mass fraction. The strongest argument against the proposed simplified relationship between sensitivity and molecular weight is actually given in figure 1: If the sensitivity would only be a function of molecular weight, why is the monomer-to-dimer ratio different between the positive and negative mode? Furthermore, we observed that ESI sensitivity between two very similar commercially available standards (pinonic acid and camphoric acid) can be different by orders of magnitude. Therefore, the authors should argue here more carefully and consider that different ionization efficiency can introduce a significant bias.

We agree that the approach we take is rather approximate and provides at best a qualitative picture of the relative abundances. This is a drastic approximation because even for a series of carboxylic acids, the ESI detection sensitivities can vary by three orders of magnitude within the same sample (Bateman et al., 2012). However, we still see value in this approach for making relative comparisons for the same types of SOA samples. We have emphasized the approximate nature of this approach in the text and added the following reference:

Bateman, A.P., Laskin, J., Laskin, A., and Nizkorodov, S.A.: Applications of high-resolution electrospray ionization mass spectrometry to measurements of average oxygen to carbon ratios in secondary organic aerosols, Environ. Sci. Technol., 46, 8315-8324, http://dx.doi.org/10.1021/es3017254, 2012.

**1.8** p.5, l.1: I agree that the signal in the lower volatility bin (Figure 3) is suppressed under high RH in the positive mode. However, for the negative mode this is not obvious. Did the authors try to normalize the figures on the base peak in the mass spectrum to present the relative changes more in detail?

We did not normalize the mass spectra. It was more important to see whether these species would be present in the gas or particle phase. See the last sentence in that paragraph copied below:

"the high RH to low RH ratio of the combined peak abundances for the compounds falling below $\log(C_0) = 1$ is 0.3 in the positive ion mode and 0.05 in the negative ion mode."

**1.9** p.6, l.22-24: Again: the authors should discuss the possibility of enhanced condensed phase hydrolysis reactions of dimers under high RH conditions. The enhanced viscosity of particles under low RH might suppress the rate of hydrolysis reactions.

We have included this possibility in last paragraph in the Results and Discussion section, and in other places in the text.

**1.10** p.13, Figure 3: The volatility distribution in (a) looks as it is cut on the high volatility end. How many products appear in the bins log10(C0)>6? Can these products be summed up into one bin which contains all compounds of all bins with log10(C0)>6?

The plot was not cut at the high volatility end. What appears as a cut was likely an outcome of the mass spectrometer limitation – it could not observes ions below m/z 100.

Technical corrections

p.13, Figure 3: The figure caption contains (a) and (b) which is not shown in the figure. Also, the colour description in the caption is wrong: high RH should be "blue" and low RH "red".

The figure caption was updated.

Literature

Krapf, M., El Haddad, I., Bruns, E. A., Molteni, U., Daellenbach, K. R., Prévôt, A. S. H., Baltensperger, U. and Dommen, J.: Labile Peroxides in Secondary Organic Aerosol, Chem, 1(4), 603–616, doi:10.1016/j.chempr.2016.09.007, 2017.

Zhao, R., Aljawhary, D., Lee, A. K. Y. and Abbatt, J. P. D.: Rapid Aqueous-Phase Photooxidation of Dimers in the α-Pinene Secondary Organic Aerosol, Environ. Sci. Technol. Lett., 4(6), 205–210, doi:10.1021/acs.estlett.7b00148, 2017.

---

## Author Comment (AC2) · 26 Dec 2017

**Response to Reviewer 2**

The Hinks et al. manuscript reports on differences in secondary organic aerosol (SOA) mass concentration and composition during high vs. low relative humidity (RH) chamber experiments. Toluene was oxidized under low-NOx and high-NOx and low (<2%) and high RH (up to 90%) conditions. Differences in mass concentration were evaluated using an SMPS; and differences in composition using nano-DESI-HRMS. While chamber experiments and SOA formation studies largely have been conducted under dry conditions, much recent attention has been given to the effects of water on particle formation and composition. Much of that effort has been focused on understanding the effects of RH on particle viscosity. There is much opportunity to advance the mechanistic understanding of SOA formation through compositional studies such as the one presented in the Hinks et al. manuscript. That said, while the methodology and results are presented clearly, the analysis and discussion could be more robust. There have been a large number of papers published on the photooxidation of toluene (and other aromatics) and subsequent SOA formation. The results of the experiments presented are not adequately placed in the context of what is already known about toluene SOA formation, including recent mechanistic studies looking at the role of aerosol liquid water on SOA formation from toluene. The results presented are new, and with some further analysis and discussion, this work could become a significant contribution to the field. This work is suitable for publication in ACP, following some strengthening of the analysis and discussion.

We felt it would be overwhelming to include all existing references on photooxidation of toluene and other aromatic compounds in this short paper. We included what we thought were the most relevant references for understanding the humidity effects. However, in response to this comment we added a few additional references, including previous toluene SOA studies and a few new papers on this topic that were submitted/published after our ACPD submission. We also made it clear in the summary  section that our conclusions do not necessarily apply to conditions when hygroscopic seed particles are present, where aerosol liquid water (ALW) can also contribute to particle growth. We hope the current version gives better justice to previous studies of toluene SOA.

Technical and Editorial Comments:

**2.1** The abstract reports that the nano-DESI-HRMS analysis was performed on filters from the low (<2%) and high (75% only) experiments, and particle size analysis on a wider range of high RH (75-90%). This is not as clear in the manuscript itself. Are the nano- DESI-HRMS results from a single experiment? Or averaged over all of the 75% RH experiments? This should be made clearer in the text and in the figures/tables.

The SMPS measurements were performed for all experiments. However, only selected samples were analyzed by HRMS. Our previous experience with HRMS analysis of SOA has shown that the mass spectra of SOA collected under the same conditions are reproducible, and we normally pick the sample with the highest signal-to-noise ratio for the analysis. We clarified it in the revised paper.

**2.2** p. 4, lines 1-4 (referencing Fig. 1/Table 1): The observation that there seems to be no mechanistic difference between high and low RH and only differences in the formation of oligomers is not completely satisfactory. For the positive ion mode, it is clear that the mass spectra are more similar (samples share the same "compounds"); it is not quite as clear that there is a reduction in peak abundance. For the negative ion mode, it is clearer that there is a reduction in peak abundance, but it is not as clear that there is similarity among the most abundant compounds. For both positive and negative mode there seems to be a reduction in peak diversity, with significant differences in the negative ion mode.

Can these differences be further probed to support the hypothesis or provide alternative hypotheses?

This is a good point. We added a statement that not only the peak intensity drops at high RH but also the observed peak number (referred here as "diversity").

One could get a deeper understanding of the toluene SOA system with a more sophisticated analysis method that separated the compounds before HRMS analysis. For the nano-DESI based approach, which is not capable of separating isobaric species the description can only be qualitative. We added a note about it as well.

**2.3** There have been a number of mechanistic/product studies of toluene, and other aromatics, under high- and low-NOx conditions.  Do those studies (e.g., observed gas-phase intermediates) combined with what is known about oligomerization pathways (e.g., as discussed on p.6) support the RH dependence under low NOx but not high NOx conditions? Citation of similar results in Cao et al. (2010) is not sufficient.

It is absolutely correct that there have been a number of studies dealing with the identity of the toluene photooxidation products. Unfortunately, almost none of them examined the chemical nature of oligomeric compounds. There is a suggestion in Sato et al. (2007) that oligomers may represent hemiacetals. We re-wrote the discussion to cite more mechanistic papers.

**2.4** p. 4, lines 5-10 (referencing Fig.  2):  While the formation of higher order oligomers is suppressed, it is interesting that the most abundant peak > C7 in the positive ion mode spectra under high RH appears to be a dimer.

Indeed, this is an interesting observation. We added a sentence pointing it out but we do not have a good chemical explanation for it.

**2.5** p. 4, lines 31-34:  The fraction of compound detected in the particle phase is discussed in terms of ambient organic aerosol (OA) levels, and an explanation is given for gas-phase adsorption artifacts.  However, based on table 2 (and discussion in the manuscript) the experimental OA levels vary widely between the low and high RH experiments. The discussion of partitioning and artifacts needs to be expanded to reflect the range of experimentally observed OA levels.  One question that arises is whether there are more artifacts with high RH than low RH and if so what may be the reason for that? There is little to no discussion of the role gas-phase chemistry plays in the observations.   There is an underlying assumption that because the initial VOC and oxidant concentrations were the same between experiments that the reacted VOC concentrations were also the same. Are there measurements to support that assumption? While many things can

influence SOA yield, were the observed yields (if known) generally consistent with other published studies?

The only gas-phase measurement carried out during these experiments was measurements of the amount of toluene reacted. Over the course of an experiment, the PTR-ToF-MS signal for toluene at low RH and at 75% RH decreased to about 60% of its original signal, at both RH values. This suggests that the rate of consumption of toluene was similar under low and high RH conditions. We have added this info to the experimental section.

We could only estimate the yield values in this study and elected not to place them in the original submission. However, based on this comment we calculated the approximate SOA yields and included them in Table 2. We have included a comparison with previously reported yields.

**2.6** Did the average particle size change between high and low NOx conditions?

Under low RH, the particle geometric mean was comparable between the high $NO_x$ (~230 nm) and low $NO_x$ (~215 nm) experiments at the time of collection; this was not the case under high RH. Under high $NO_x$ conditions, the particle size remained constant across the various RHs. In contrast, under low $NO_x$ conditions, the particle geometric mean decreased as RH increased.

**2.7** p. 5, lines 17-27:  The authors mention the hygroscopicity of SOA here, but further discussion may be helpful. Is the amount of aerosol liquid water under high RH (75%) sufficient to prohibit/limit condensation reactions?

Aerosol liquid water is not needed for the condensation reactions since it is a product. In fact, in organic synthesis, condensation reactions go better when a desiccant is added to the mixture to remove the water from it as it forms. In the aerosol experiments such as the ones described here dry air acts as the desiccant.

Since SOA is not too hygroscopic water would be present in particles in trace levels. Even under humidified conditions, it is probably not sufficient for hydrolysis reactions in the particle (we do mention it as a possibility because reviewer #1 insisted on including hydrolysis as a possible explanation).

**2.8** p. 7, lines 3-12: The extension of the experimental observations to atmospheric implications is not well supported and unnecessary. There are so many elements of experimental design that may affect the results and the extension to the ambient atmosphere; these include absence of seed aerosol (chemical and physical effects), absolute levels of precursors and particles, and relative levels of radicals. Further in the ambient atmosphere the gas-phase chemistry is controlled by more than a single VOC precursor, and the particle composition will affect the extent of aerosol liquid water.

We removed the two paragraphs discussing possible implications of the measurements. However, we kept the paragraph suggesting that RH dependence of SOA from other aromatic compounds should also be investigated. We also added a paragraph that compares the seeded and seedless experiments, and stated that the effect of the seed is likely stronger than the effect on the oligomers observed in this study.

**2.9** Figures: It is recommended to specify the NOx and RH conditions in the figure captions just as a reminder since high-NOx and a range of RH are discussed in the manuscript

The figure captions were updated.

---

## Author Response (AR2)

Comments to the Author:

Please correct the following typos:

p8 l4: "an" opposite "effect"

p8 l5: "to the aqueous"

p8 l5: and "subsequent"

p8 l9: "enhancement"

All these typos were correct. In addition, we read the paper several more times and fixed a few additional typos, including cleaning up the reference section. Our tracked changes are shown on the next page.

For future submissions: when providing a revised manuscript, please highlight all changes so that the changes can easily be spotted by the editor or reviewers.

We apologize for not better drawing your attention to the changes we made in the manuscript. We attached the track-changed version of the paper to the responses to review comments, and a separate clean version of the paper, as ACP instructions suggested. We assumed that the track changes should have made it clear what was deleted and what was inserted.

[revised manuscript text omitted]